# Microcystin Concentrations, Partitioning, and Structural Composition during Active Growth and Decline: A Laboratory Study

**DOI:** 10.3390/toxins15120684

**Published:** 2023-12-06

**Authors:** Emily F. Pierce, Astrid Schnetzer

**Affiliations:** Department of Marine, Earth and Atmospheric Sciences, North Carolina State University, Raleigh, NC 27695, USA; epierce@ncsu.edu

**Keywords:** microcystin, cyanobacterial harmful algal bloom, microcystin congeners, microcystin persistence, particulate and dissolved phases, North Carolina

## Abstract

Microcystin can be present in variable concentrations, phases (dissolved and particulate), and structural forms (congeners), all which impact the toxicity and persistence of the algal metabolite. Conducting incubation experiments with six bloom assemblages collected from the Chowan River, North Carolina, we assessed microcystin dynamics during active growth and biomass degradation. Upon collection, average particulate and dissolved microcystin ranged between 0.2 and 993 µg L^−1^ and 0.5 and 3.6 µg L^−1^, respectively. The presence of congeners MC-LA, -LR, -RR, and -YR was confirmed with MC-RR and MC-LR being the most prevalent. Congener composition shifted over time and varied between dissolved and particulate phases. Particulate microcystin exponentially declined in five of six incubations with an average half-life of 10.2 ± 3.7 days, while dissolved microcystin remained detectable until the end of the incubation trials (up to 100 days). Our findings suggest that concerns about food-web transfer via intracellular toxins seem most warranted within the first few weeks of the bloom peak, while dissolved toxins linger for several months in the aftermath of the event. Also, it was indicated there were differences in congener profiles linked to the sampling method. We believe this study can inform monitoring strategies and aid microcystin-exposure risk assessments for cyanobacterial blooms.

## 1. Introduction

Microcystins (MCs) are a group of cyclic hepatotoxins produced by different genera of cyanobacteria including *Microcystis*, *Dolichospermum*, *Planktothrix*, and *Nostoc* [1,2]. MCs are the most commonly occurring cyanotoxins globally and have been investigated in various types of waterbodies, especially rivers, lakes, and estuaries prone to cyanobacterial harmful algal blooms (cHABs) [3,4,5,6]. MCs adversely affect livestock and wildlife and have been linked to gastrointestinal symptoms, liver cancer, and death in humans [7,8,9,10,11]. The most common pathways for exposure are through recreation (e.g., swimming) and ingestion which may occur when contaminated fish, shellfish, or drinking water are consumed [12]. Toxic cHABs can result in significant economic losses due to recreational area closures, fishery shutdowns, or drinking-water contamination [10,13,14]. As climate change in combination with eutrophication is expected to result in increases in the frequency and magnitude of cHABs, the development and implementation of protective guidelines has become a major focus for water-quality managers and stakeholders [15,16].

The World Health Organization (WHO) has published a set of guidelines for MC-related human health risks based on proxy measurements including cyanobacteria cell densities (cells mL^−1^) and chlorophyll-a (chl-a) concentration (µg L^−1^), with low risk associated with up to 20,000 cells mL^−1^and 10 µg L^−1^, respectively, moderate risk associated with between 20,000 and 100,000 cells mL^−1^ and 10 and 50 µg L^−1^, respectively, and high risk associated with greater than 100,000 cells mL^−1^ and 50 µg L^−1^, respectively [17]. While these metrics help management agencies assess general bloom dynamics (e.g., biomass changes from initiation to peak and demise) at a relatively low cost, they may not be accurate proxies for toxin dynamics during these events [16]. For instance, not all cyanobacteria are capable of toxin production, and those that are may not produce toxins continually [18]. In addition to chl-a- and cell-density-based guidelines, the WHO also established thresholds based on MC concentration at 1 µg MC L^−1^ for drinking water and 10 µg L^−1^ for recreational uses [17]. An important caveat to the WHO guidelines is that they are based solely on microcystin-LR (MC-LR), which is only one of more than 200 structural variants also known as congeners [19,20]. An increasing number of studies indicate that cHABs can be associated with multiple congeners besides MC-LR, commonly including MC-LA, MC-RR, and MC-YR to name a few, and that these congeners vary in their persistence and toxicity [21,22,23,24]. For example, MC-LR has been shown as being more toxic than some other MC congeners based on animal models [25] but has also been observed to degrade faster than other variants [24,26]. These findings highlight the need for a more comprehensive analysis of toxin structural profiles over varying bloom stages.

In addition to the potential shifting in MC structural profiles during a bloom, there is also variability in how MC partitions between particulate (intracellular, pMC) and dissolved (dMC) fractions, with the latter being released into the surrounding water. While WHO guidelines for total MC (pMC + dMC) do not distinguish between these fractions, they play a key role in assessing exposure pathways. PMC is typically associated with food-web contamination, while dMC is more likely to cause an issue for drinking-water resources. It has been posited that MCs remain in the particulate phase until cell death and that very little dMC will be present during a bloom [27,28]. However, dMC has been observed to make up greater than 25% of total MCs in multiple observational studies [24,29]. Toxin partitioning between phases may also impact toxin persistence with dMC-LR taking twice the time than its particulate equivalent to decrease by 90% [30]. These findings corroborate that information on toxin partitioning over varying bloom phases is an integral piece to gauging the severity and length of MC exposure risks.

For this study, a series of algal growth and degradation experiments were conducted to document changes in MC concentration, partitioning, and structure using six bloom assemblages collected from the Chowan River, a major tributary within the Pamlico-Albemarle Sound System, North Carolina (NC). The main aims of this work are to assess how long MCs may persist throughout a bloom, if toxin persistence varies across phases, and if congener profiles change across phases or throughout bloom stages. Investigating temporal patterns in toxin congener profiles and phase partitioning can provide key insights into bloom toxicity and persistence, and thus can aid management entities in assessing health risks during and after bloom events. 

## 2. Results

### 2.1. Chlorophyll-a and Microcystin Concentrations

Chlorophyll-a (chl-a) levels ranged from 120 to 2772 µg L^−1^ upon collection of the assemblages. After initial nutrient amendments to each of the communities at the beginning of the incubations, the assemblages showed differing growth responses with average chl-a levels peaking at 14,105 µg L^−1^ for the assemblage from Colerain (CR), 1023 µg L^−1^ for Arrowhead (AH), 2141µg L^−1^ for Indian River (IR), 2013 µg L^−1^ for Leary’s Landing (LL), 4774 µg L^−1^ for Modoc Canal (MC), and 1680 µg L^−1^ for Charlton Pier (CP) (Appendix A). Chl-a concentrations differed by site (*p* < 2 × 10^−16^, ANOVA) and by the interaction term between the site and day of bloom collection (*p* = 0.04, ANOVA). 

Initial average and maximum average pMC concentrations ranged from 0.2 µg L^−1^ to 993 µg L^−1^ and 0.45 µg L^−1^ to 993 µg L^−1^, respectively, across all incubations (Figure 1). Maximum pMC declined to less than half after 10 days at every bloom site and dropped to less than 10% by day 40, with the exception of CP which had a much lower pMC concentration at the beginning of the experiment. A decline in pMC concentrations occurred in all incubations, regardless of initial pMC or chl-a concentrations (Figure 1).

The rate of exponential loss of pMC and the fit of the exponential-loss model varied between experiments (Figure 2; Table 1). The exponential-loss models explained 50% or greater of the variation in pMC concentration over time for all but the CP experiment (Table 1). The modeled rate of pMC decline ranged between 0.04 and 0.12 day^−1^ (excluding CP).

### 2.2. Dissolved Microcystin Concentrations

Initial average and maximum average dMC concentrations ranged from 0.5 µg L^−1^ to 3.6 µg L^−1^ and 1.3 µg L^−1^ to 75.6 µg L^−1^, respectively, across all incubations (Figure 3). A net increase in dMC concentration during the incubation period was observed in each experiment, but changes over time were variable across the different incubations. 

### 2.3. Microcystin Structural Profiles

The initial structural profile of pMC was consistent across assemblages with MC-RR as the most abundant congener, followed by MC-LR, MC-YR, and MC-LA (Table 2). 

For the two experiments with additional congener data from later in the incubations, CR and LL, there were variations in the structural profile between phases (dissolved versus particulate) and within phases over time (Figure 4). MC-RR contributed the most to the particulate pools at both sites except for the last time point for CR when MC-LR increased its relative contribution. MC-LR also exceeded the contributions from other congeners in the accumulated dissolved pool at both sites. In contrast to the CR site, MC-LR contributed the most to the dissolved fraction at site LL except for the last time point when MC-RR and MC-YR equally made up the congener profile.

The average relative abundance across phases of MC-LR increased over time (*p* = 0.05, ANOVA), and the average relative abundance across phases of MC-RR decreased over time (*p* = 0.01, ANOVA) in experiment CR but not experiment LL. The contributions from MC-LR, MC-RR, and MC-YR also differed when comparing dissolved, accumulated dissolved, and particulate fractions at both sites (Table 3).

### 2.4. Comparing MC Concentrations from ELISA and LC-MS

There was generally good agreement between LC-MS and ELISA MC quantification for particulate and SPATT concentrations (Figure 5; y = 1.23x, *p* = 7 × 10^−16^; y = 1.03x, *p* = 3 × 10^−7^). For discrete dissolved samples, however, ELISA estimates resulted in higher MC concentrations compared to LC-MS (y = 0.06x, *p* = 0.03).

## 3. Discussion

### 3.1. Shifts in Microcystin Concentration and Partitioning

Building an understanding of MC dynamics throughout bloom progression, especially for specific MC phases (pMC and dMC) and congeners, has become a major focus of ongoing cHAB research to better inform bloom monitoring strategies and assess exposure pathways and risk [31]. In this study, an exponential decline in pMC in five out of the six experiments led to a half-life of 10.2 ± 3.7 days. Despite this seemingly fast release, due to the high initial pMC concentrations that ranged from 37 to 993 µg L^−1^, average pMC did not fall below the recreational guidance level of 10 µg L^−1^ [17] until day 50, with a considerable range from 14 to 96 days. Partially, these differences in experimental growth and toxin dynamics, including half-time estimates, were attributed to the retrieval of assemblages from the Chowan River during varying bloom stages (e.g., likely near to the bloom peak for CR versus the declining phase for CP). More indication that bloom status varied across the assemblages upon collection was that an initial nutrient amendment only triggered additional increases in pMC concentration in a subset of communities from AH, LL, and MC within the first 3–5 days before exponential toxin decline began. The stimulation of additional growth, which was indicated by changes in chl-a in several of the experiments, did not link to shifts in pMC levels except for short-lived, small increases in IR and CP. In line with the findings from other MC production studies, cellular resources may have been diverted away from MC production and toward growth due to limiting conditions further explaining a lack of pMC production during later time points [32]. 

A limited number of studies report on pMC decline using natural assemblages with pMC half-lives similar to the estimates in this study (Table 4). A direct comparison, however, requires caution due to differences in methodology including the use of amendments (i.e., nutrients, algicides, or specific congeners), temperature and light conditions, the makeup of the algal assemblage, the observational approach (bottle, mesocosm, in situ sampling), MC detection method (HPLC, LCMS, ELISA), and, to name one more, whether the half-life was based on a single congener or a combined congener signal (Table 4). Studies that did not add algicides or congeners but based their observations on already present toxins included a mesocosm trial in Lake Tuusulanjärvi in southern Finland during the summer and fall months where pMC half-lives were modeled based on MC-LR with 4–5 days (Table 4) [30]. Another study in a small lake in Ontario, Canada, following a *Microcystis* bloom yielded half-lives of ~7 days for MC-LA in situ, while incubating the same assemblage under varying laboratory conditions (e.g., decreased light and decreased temperatures) significantly increased estimates to 24–55 days (Table 4) [24]. This later study highlights how shorter pMC half-lives from in situ studies are likely linked to abiotic and biotic natural conditions that exacerbate the breakdown and release of pMCs [24,33]. In addition to physicochemical factors, precipitation, turbulence, and flow may also contribute to pMC breakdown [34]. For example, during a study of MC dynamics following the algicide treatment of an Australian lake, a flood occurred that washed out and diluted all MC from the study site (Table 4) [35]. 

Finally, the importance of potential MC-degrading bacteria as proxies for bloom stage and toxicity have been discussed previously in relation to pMC release [26,32,36]. While no community data were collected to characterize any shifts in the assemblages during the incubation trials, the initial characterization of the bloom-associated bacteria using high-throughput sequencing highlighted different relative contributions from taxa that have been associated with MC degradation and also indicated that Microcystis was the dominant toxin-producing genus upon bloom collections [37]. Based on experimental half-lives from this study, the risks of pMC exposure via algal biomass consumption and food-web transfer lasted for several weeks depending on uptake rates and the accumulation potential of consumers (e.g., varying shellfish or fish) [38,39]. However, in situ, this risk would likely diminish quicker due to varying factors including accelerated pMC release in response to changing environmental conditions (e.g., temperature or mixing), microbial degradation stimulated by additional resident populations, or the presence of cyanophages and predatory zooplankton [24,30,40,41,42]. Iterative field and laboratory studies are needed to decipher how these complex drivers affect bloom progression and toxin release from cells and if the latter may be linked to congener composition (Table 4) [22].

A general increase in dMC concentrations throughout most of the incubations except for the CP experiment was observed but not immediately apparent since dMC was also removed from the dissolved phase by adsorption to the duplicate SPATTs in each bottle. Maximum dMC recorded in the water and accumulated in SPATTs did not fully account for the total concentration of pMC released from cells up to the time point when the second SPATT was removed (average range = 3.2–28.8%, *n* = 5), indicating that some fraction of dMC was degraded. Dissolved MC remained above safe drinking-water thresholds (1 µg L^−1^ [17]) up to 100 days after the start of the experiment and was thus persistent, which was in agreement with lake studies that determined shorter pMC than dMC half-lives (Table 4) [24,30]. While dMC has been detected following a bloom for lengthy time periods, experiments utilizing added MC spike-ins to natural assemblages found that the bacterial degradation of dMC-LR and dMC-RR resulted in differing, congener-specific half-life estimates of 0.06–8.3 and 1.1 ± 0.23 days, respectively (Table 4) [32,43]. Spiking lake water with dMC and following bacterial degradation indicated that congeners MC-LR and MC-LF also differ in their persistence [26]. Interestingly, this study also linked the degradation of dMC to water “history”, where water from locations previously exposed to toxic events led to a faster breakdown. A follow-up study using water from the same locations linked degradation capacity to differences in the resident bacterial composition [36]. The juxtaposition of degradation and the long-term persistence of dMC in this study and others exemplifies the need for more research into dMC degradation rates and pathways. The persistence of dMC over potentially lengthy periods of time, even spanning seasons, could have significant implications for post-bloom management. Dissolved MC is a primary concern for drinking water and while dMC was not a primary concern for recreational exposure, there is research suggesting transdermal uptake may be a more important exposure route than currently reflected by health and safety guidelines [44].
toxins-15-00684-t004_Table 4Table 4Review of the published degradation rates of pMC and dMC under varying conditions.Study TypeInitial Amendment or ManipulationDominant Organism in Sample Community ^a^Method*n*Temperature (°C)Light Conditions ^b^CongenerpMC Half-Life(Days)dMC Half-Life(Days)Ref.In vitroNutrients*Microcystis*ELISA ^c^153375 μE m^−2^ s^−1^-RR, -LR, -YR, -LA5.8–17.3-This studyIn vitro-*Microcystis*HPLC225260 μE m^−2^ s^−1^-LA44.9 ± 0.763.5 ± 5.3[24]



22545 μE m^−2^ s^−1^42.8 ± 0.7120.4 ± 1.0



225Dark23.8 ± 2.4131.5 ± 7.5



24Dark54.6 ± 0.5251.0 ± 35.9In situ-*Microcystis*HPLC5--6.5 ± 0.415.8 ± 1.0In situ-*Microcystis wesenbergii*HPLC1AmbientAmbient-LR4.710.0[30]MesocosmUncovered2AmbientAmbient4.310.2
Covered2AmbientDark4.38.9Studies following algicide treatmentIn vitroCopper*Microcystis*HPLC225260 μE m^−2^ s^−1^-LA9.2 ± 0.710.9 ± 0.3[24]

22545 μE m^−2^ s^−1^10.5 ± 0.926.5 ± 0.9

225Dark5.0 ± 0.133.8 ± 2.2

24Dark24.2 ± 1.331.3 ± 1.8In situCopper5--1.5 ± 0.032.8 ± 0.3In vitroCopper sulfate or lime*Microcystis aeruginosa*HPLC220 ± 2900 lux-LR-3[45]In situCopper*Microcystis aeruginosa*HPLC2-Ambient-LR-1–5[35]Studies following dMC amendmentIn vitroMC from lysed algal material-HPLC120 ± 2Dark-LR-<4[46]In vitroMC from purified standard-HPLC217 ± 0.5Ambient-LR-3–4[47]In vitroMesocosmMC spike-in lysed algae or purified standard*Microcystis*ELISA ^c^320Dark-LR-0.9 ± 0.07[43]920Dark-LR, -RR-1.16 ± 0.112AmbientAmbient-LR-0.66 ± 0.112AmbientAmbient-RR-1.1 ± 0.23In vitroMC from purified standard-HPLC1829--LR-4–14[26]929--LF-9–22In vitro_15_N-MC*Microcystis*LC-MS3Ambient300 μE m^−2^ s^−1^-LR-0.08–6.3[32]^a^ All studies used natural assemblages from lakes or reservoirs. ^b^ All studies stated light levels were applied in natural or 12:12 h cycles. ^c^ HLPC or LC-MS used to confirm congener composition but ELISA used for MC quantification.

### 3.2. Microcystin Congener Profile

Differences in MC structural composition between bloom events or a shift over time during bloom progression, as observed in this study, have significant implications for toxin-exposure risk management due to varying congener toxicity [24,48,49,50]. Nevertheless, environmental testing and human toxicity studies have primarily focused on MC-LR, and guidelines for drinking, recreation, and consumption are developed based on this single congener [17]. Moreover, toxin testing typically results in measurements of total MC without a further distinction between particulate and dissolved phases. For the two experiments (CR and LL) for which congener profile changes were analyzed, MC-LR and MC-RR were the most abundant congeners overall, which is in agreement with a previous metanalysis of congener composition in US lakes [51]. In addition to the dominant derivates, MC-YR and MC-LA were present in much smaller abundance despite MC-LA showing more prevalence in North American waterbodies elsewhere [52,53]. The two dominant structural forms in this study necessitate different management responses given that MC-LR is roughly ten times as toxic as MC-RR [9,25,48]. Moreover, as the average relative contribution of MC-LR increased and that of MC-RR decreased, the CR bloom may have also shifted toward a more toxic composition overtime. Such a shift, as well as the potential synergistic effects of varying congeners, would need to be considered to effectively understand risk severity. Also detected was a notable difference in congener contribution between phases as MC-YR made up significantly more of the dMC compared to the pMC pool in the LL assemblage compared to CR, which may suggest a higher persistence of the congener. This notion, however, does not agree with previous experiments that found dMC-YR to be more labile than dMC-LR using purified MC amendments of MC-YR and MC-LR to non-axenic *M. aeruginosa* cultures (24 °C, dark) [50]. As congener persistence may not be an inherent property, major knowledge gaps on how differential congener composition profiles may link to cyanobacterial community composition and how they may relate to environmental forcings will help to better understand congener turnover and changes in exposure risk [53].

This study corroborates that the method of toxin sampling can be of key importance for toxin characterization. For instance, the more toxic MC-LR congener contributed more strongly to the dMC pool based on SPATT analyses compared to discrete dMC sampling which represented a snapshot of dMC concentrations, whereas SPATT profiles were derived from accumulation of congeners over time. While adsorption efficiencies have been shown to be nearly identical across all four tested congeners from this study, Kudela et al. [54] also indicated that the MC-RR congener may be extracted at lower efficiencies from SPATT resin, potentially altering the yielded profile. So, although SPATTs are low-cost, in situ monitoring tools that have been widely employed, especially in environments where no prior data exist on toxin presence, it has yet to be assessed how well the SPATT methodology facilitates the accurate tracking of MC congener contributions.

Finally, there was a strong agreement in MC concentrations based on the congener-specific LC-MS and congener-indiscriminate ELISA approaches for the particulate and accumulated dissolved phases. Given that the ELISA used in this study was sensitive to the ADDA side chain which is present in most structural MC variants [34,35], the agreement between the approaches supported that MC-LR, -RR, and -YR were indeed major MC contributors. Lake Erie studies have shown the dominance of these same congeners in association with *Microcystis* spp. blooms [55,56]. A discrepancy between ELISA and LC-MS-derived concentrations for the discrete dMC phase, driven by samples collected after the initial time point and during decomposition in the CR experiment, was likely indicative of congeners other than the four we tested and possibly linked to transformational changes to congeners released from the cells [57]. As the ELISA approach offers relatively quick estimations of MC concentrations including most structural forms, its complementation with newly developed congener-specific LC-MS methods will continue to provide vital insights into MC dynamics [55,58,59,60,61].

## 4. Conclusions

This study provides several takeaways that can inform monitoring strategies and bloom management. First, pMC is released from cells at exponential rates independent of chl-a or overall maximum MC concentrations, indicating that toxin release is not correlated with bloom severity or overall biomass decline. Second, dMC is more persistent than pMC for naturally occurring congener mixtures. Dissolved MC may persist well after pMC disappears and chl-a begins to decline, and thus bloom management should incorporate testing requirements for recalling recreational or drinking-water closures. Third, MC congener profiles shifted over time and between phases and were also distinct between the incubations of bloom assemblages from two events. Changes in congener composition confer changes in toxicity and toxin persistence in dissolved and particulate phases, and thus congener assessment should be a part of comprehensive bloom management to mitigate toxin exposure risks. Potential future work on this subject should further investigate how differences in environmental conditions or the bacterial community may relate to differential MC degradation.

## 5. Materials and Methods

### 5.1. Sample Collection and Experimental Setup

Natural bloom assemblages were collected from the surface (0–0.5 m) using polycarbonate carboys or polyethylene terephthalate glycol (PETG) sampling bottles, allowing them to fill slowly, and then the water was transported to the laboratory where the assemblages were kept at the ambient temperature they were collected at (~33 °C) under a light:dark (L:D) cycle of 12:12 h at 75 μE m^−2^ s^−1^ using cool white fluorescent lights until experimental setup within 24 h. The collection dates ranged from July to September of 2019, and the 6 sites were located along the Chowan River (Appendix A; Appendix A). The microbial assemblages were incubated as triplicates using 1L PETG bottles, amended with F/2 growth medium [62] at the onset of the incubation and kept at the same aforementioned temperature and light conditions for 8–14 weeks. Samples were collected from each bottle to measure chl-a, pMC, and dMC concentrations at the beginning of the incubations (T0) and subsequently every other day for the first week, then weekly until the end of the first month, and then biweekly until the end of the experiment in the third month. In addition to sampling at these discrete time points, accumulated dMC concentrations were also determined using solid phase adsorption toxin tracking (SPATT) by immersing 2 SPATT units in each bottle and retrieving one around weeks 4 and 8.

### 5.2. Chlorophyll-a Analysis

Approximately 1–10 mL of water from each sample bottle was filtered onto 0.7 µm Whatman GF/Fs, and the filters stored at −20 °C until analysis. Filters were thawed and suspended in 7 mL of 100% acetone, sonicated for 5 s at 50% intensity (Fisher Scientific, Hampton, NH, USA, Model 120 Sonic Dismembrator), and extracted in the dark at −20 °C for 24 h [63]. Samples were run fluorometrically (Turner Designs Trilogy Laboratory Fluorometer) using the non-acidification method detailed by Welschmeyer [64]. 

### 5.3. Total Microcystin Analysis

Samples (1–10 mL) were filtered onto 0.7 µm GF/Fs (Whatman grade, GE Healthcare Life Sciences, Chicago, IL, USA) to determine pMCs and 1.5 mL of filtrate collected in 2 mL glass autosampler vials (Thermo Fisher Scientific, Hampton, NH, USA) to measure dMCs. Both filters and filtrate were stored at −20 °C until analysis. Filters were extracted through one freeze/thaw cycle in 3 mL of Milli-Q water followed by a 30 s sonication at 50% intensity (Thermo Fisher Scientific, Hampton, NH, USA, Model 120 Sonic Dismembrator) [65]. SPATT extractions followed previously published protocols [54], except elutes were combined into one sample prior to further processing. All toxin samples were analyzed using MC-ADDA ELISA kits (Product #520011, Golden Standard Diagnostics, Westminster, PA, USA). It is important to note that this ELISA kit is sensitive to MC-LR, -YR, -LF, -RR, -LW, and nodularin, and yields a combined signal for all these derivates herein referenced as total MCs. ELISA microplates were read at 450 nm using a BioTek ELx800 Absorbance Microplate Reader (BioTek, Winooski, VT, USA). 

To estimate the amount of dMC adsorbed into each SPATT unit, the SPATT extraction concentrations (ng g resin^−1^) were multiplied by the weight of HP20 resin in the SPATT (~3 g) and then normalized to µg L^−1^ using the incubation bottle volume (400 mL). Additionally, to account for the overlap in SPATT bag deployments within each incubation bottle, the dMC concentration from the first retrieved SPATT was doubled to account for dMC adsorbed by the second SPATT for the first retrieval. Subsequently, the dMC concentration from the first retrieved SPATT was added to the accumulated signal of the second SPATT tracker for the second SPATT retrieval time point. Finally, for both retrieval time points, the discrete dMC concentration at the two removal points was also added (µg L^−1^) to yield a snapshot of dissolved toxin present at the SPATT removal time points (Figure 3).

### 5.4. Microcystin Congener Analysis

In addition to deriving a combined signal for MC concentrations using ELISAs, we also obtained congener-specific information for two of the bloom assemblages from CR and LL. Congener profiles were analyzed for pMC and dMC samples at T0 and on day 26 and 58. These latter time points coincided with the removal of SPATT devices during these two experiments. T0 concentrations for chl-a indicated the CR and LL assemblages represented the most toxic events sampled. Filters, filtrate, and SPATT extracts were therefore selected for liquid chromatography mass spectrometry (LC-MS) to derive information on the presence and concentration of congeners: MC-RR, -LR, -LA, and -YA. For later comparison between total MCs derived from ELISAs, it is important to note that the sum of LC-MS-measured derivates do not entirely match those from ELISAs which combine MCY-LR, -YR, -LF, -RR, -LW, and nodularin. All LC-MS analysis was conducted by the Molecular Education, Technology and Research Innovation Center at North Carolina State University using an Orbitrap Exploris 480 mass spectrometer (Thermo Fisher Scientific, Hampton, NH, USA). For pMC structural composition, 1–10 mL of sample from each bottle was filtered onto 0.7 µm Whatman GF/Fs and extracted in an 80% MeOH solution for 12–24 h, dried in vacuo, and resuspended in a 50% MeCN solution of varying volume dependent on the expected MC concentration for LC-MS analysis. For dMC and accumulated dMC structural composition, 0.2 mL of filtrate and 0.5 mL of SPATT extract, respectively, were dried in vacuo and resuspended in 50 µL of 50% MeOH for LC-MS analysis (METRIC, pers. communication).

### 5.5. Statistical Analysis

The statistical analysis for this study was completed using Rstudio [66]. Data visualizations were produced using the ggplot2 package [67]. An ANOVA test within the AICcmodavg package [68] was used to assess for differences in chl-a concentrations, MC concentrations, and MC congener profiles. Pairwise differences from the ANOVA were determined using the TukeyHSD function in the stats package [69]. An exponential decay model was fit for each assemblage to model the decline in pMC concentrations over time: pMC Conc. (µg L^−1^) = a x e^−b × Day^(1)

Exponential decay models were fit as nonlinear models using the nls function from the stats package [69]. Linear models with a forced zero intercept were fit between MC concentrations derived from ELISA methods and MC concentrations derived from LC-MC methods for each form of MC measured (particulate, dissolved, and accumulated dissolved from SPATTs) using the lm function from the stats package [69].

## Figures and Tables

**Figure 1 toxins-15-00684-f001:**
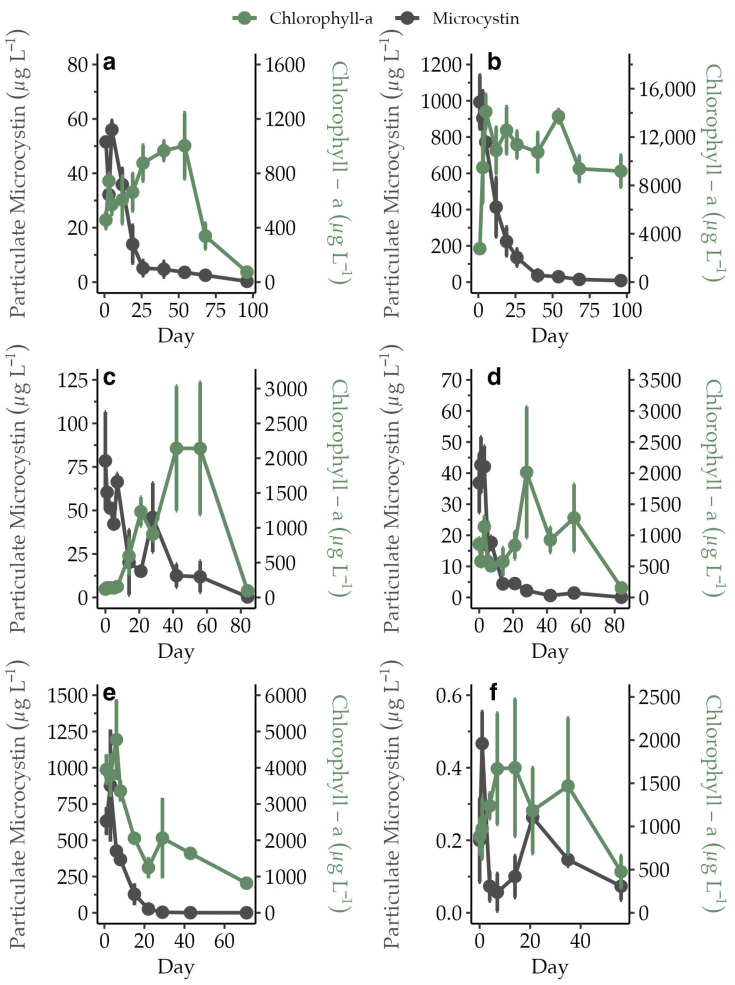
Average pMC concentration (µg L^−1^, *n* = 3) on the primary *y*-axis and chl-a concentration (µg L^−1^, *n* = 3) on the secondary *y*-axis over time for experiment AH (**a**), CR (**b**), IR (**c**), LL (**d**), MC (**e**), and CP (**f**). The standard error is shown as error bars. Note the differences in scale for the primary and secondary *y* axes.

**Figure 2 toxins-15-00684-f002:**
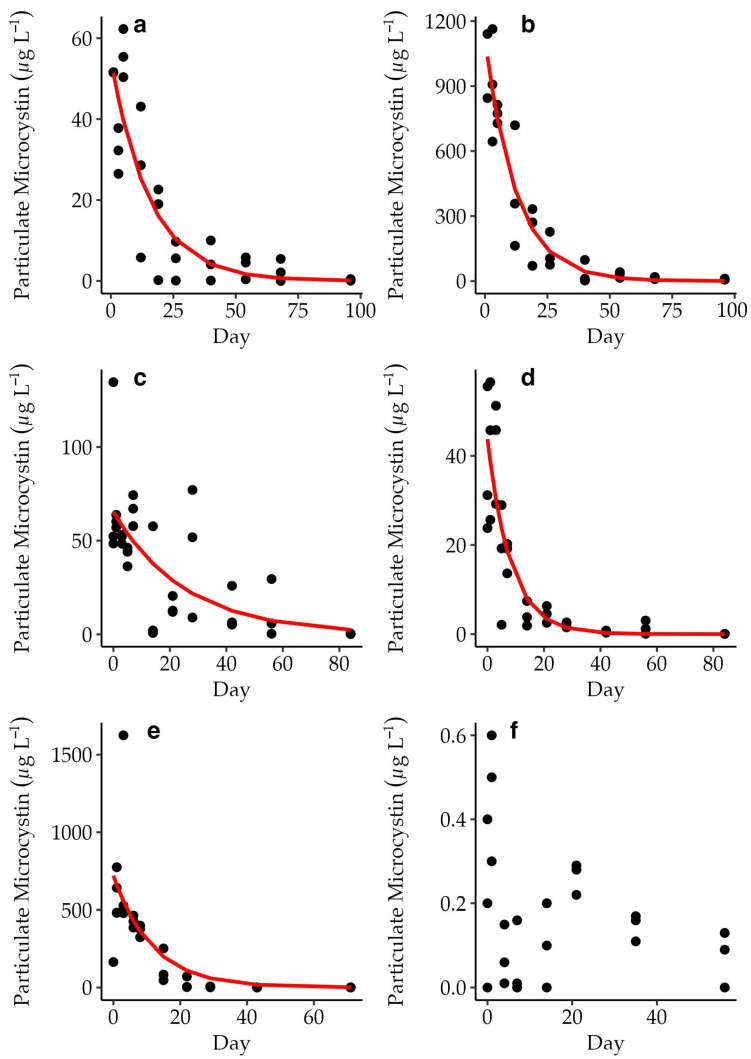
pMC concentrations from individual incubation bottles in each experiment (µg L^−1^) shown over time. Changes in pMC followed an exponential decline for AH (*n* = 29) (**a**), CR (*n* = 29) (**b**), IR (*n* = 33) (**c**), LL (*n* = 33) (**d**), and MC *(n* = 28) (**e**), but not CP (*n* = 24) (**f**). The exponential-loss model is plotted in red (see also Table 1). Note the differences in scale for the primary *y*-axes.

**Figure 3 toxins-15-00684-f003:**
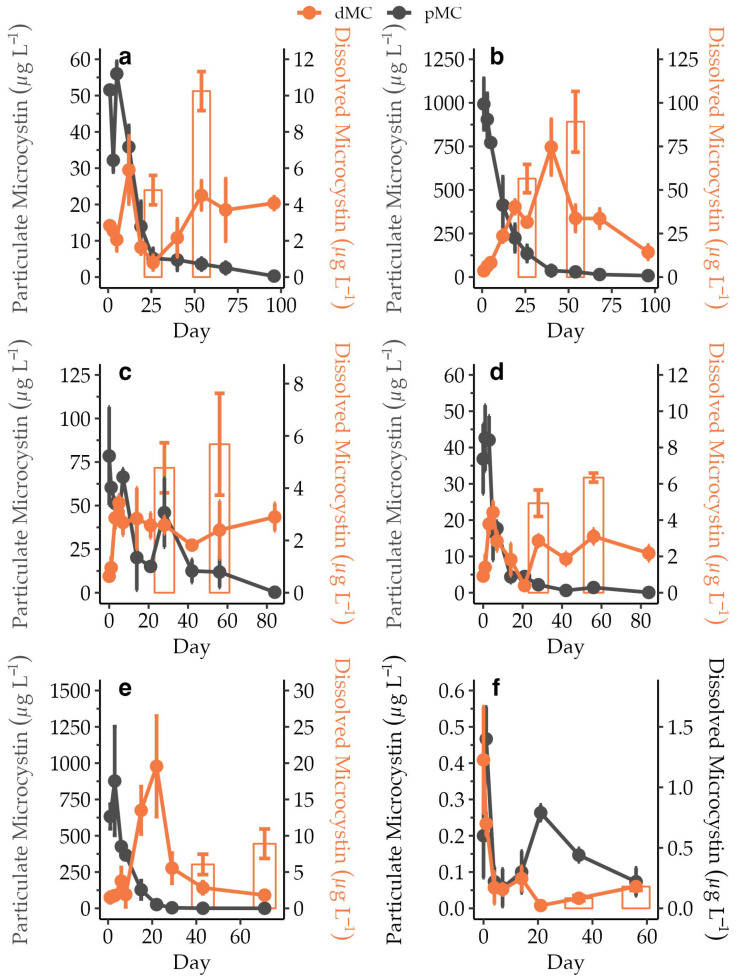
Average pMC concentration (µg L^−1^, *n* = 3) on the primary *y*-axis and average dMC concentration (µg L^−1^, *n* = 3) on the secondary *y*-axis over time with their standard errors. Also depicted on the secondary *y*-axes are the accumulated average dMC levels (µg L^−1^, *n* = 3) based on SPATT signals. Each bar shows a snapshot of the total accumulated dMC at each of the removal points after correction for the removal of dMC by the second SPATT (see further details in the text). Panels represent the dynamics for AH (**a**), CR (**b**), IR (**c**), LL (**d**), MC (**e**), and CP (**f**). Note the differences in scale for the primary and secondary *y*-axes.

**Figure 4 toxins-15-00684-f004:**
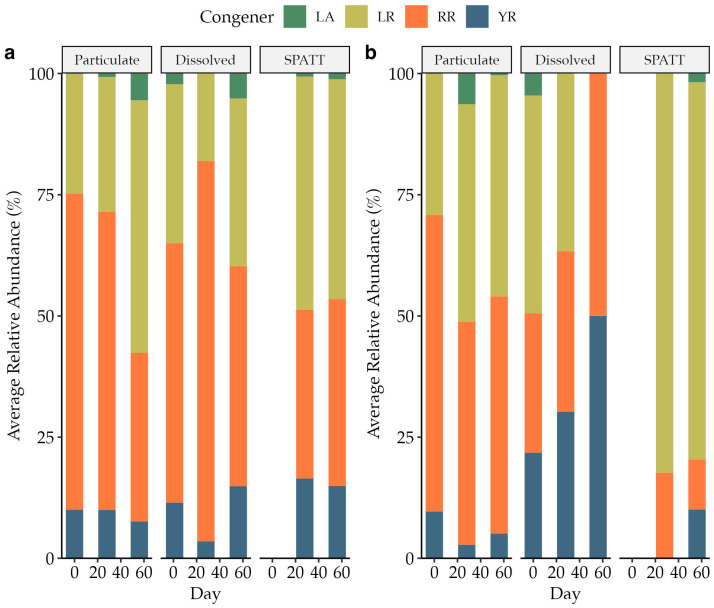
Average relative abundance of measured congeners (MC-LA, -LR, -RR, -YR) in each phase (particulate, dissolved, SPATT) for experiments CR (**a**) and LL (**b**).

**Figure 5 toxins-15-00684-f005:**
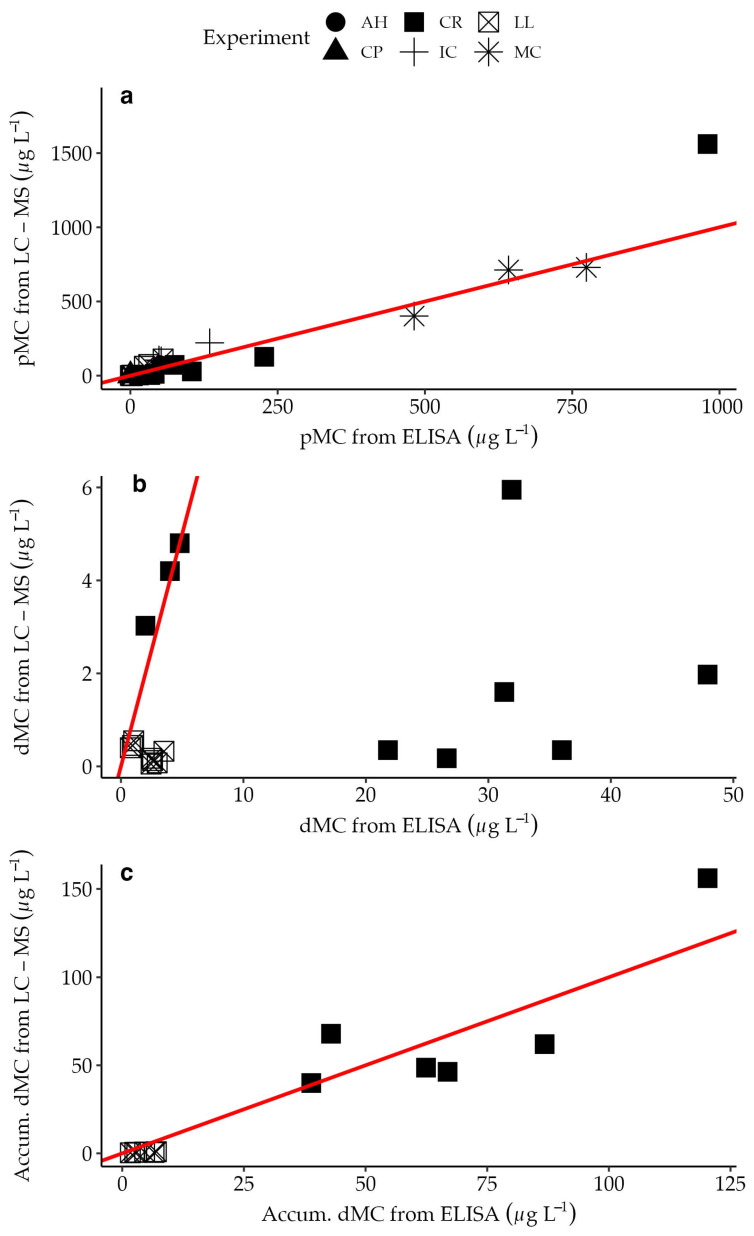
MC concentrations determined using the LC-MS method plotted against MC concentrations determined using ELISA for pMC (**a**), dMC (**b**), and accumulated dMC from SPATT (**c**). Different sites are represented by different shapes. The red line represents a 1:1 match between the two methods.

**Table 1 toxins-15-00684-t001:** Coefficients and R^2^ values of the exponential-loss model for each experiment.

	Half-Life	a	b	R^2^
Arrowhead (AH)	11.6	54.8	0.06	0.8
Colerain (CR)	8.7	1120.0	0.08	0.9
Indian Creek (IC)	17.3	65.4	0.04	0.5
Leary’s Landing (LL)	5.8	43.8	0.12	0.8
Modoc Canal (MC)	7.7	720.0	0.09	0.5
Charlton’s Pier (CP)	-	-	-	0.1

**Table 2 toxins-15-00684-t002:** Absolute congener concentrations (µg L^−1^) within the particulate phase for each experiment at the time of collection.

	MC-RR	MC-LR	MC-YR	MC-LA
AH (*n* = 1)	35.98	29.69	3.49	0.60
CR (*n* = 3)	430.50 ± 23.90	1139.77 ± 96.50	175.50 ± 16.00	1.48 ± 0.170
IC (*n* = 3)	70.30 ± 21.50	62.90 ± 31.40	8.30 ± 3.30	0.20 ± 0.06
LL (*n* = 3)	51.00 ± 12.50	24.3 ± 5.90	8.10 ± 2.30	0.05 ± 0.00
MC (*n* = 3)	347.00 ± 83.20	236.50 ± 67.40	30.10 ± 11.80	0.83 ± 0.28
CP (*n* = 3)	0.20 ± 0.06	0.13 ± 0.03	0.02 ± 0.03	0.01 ± 0.01

**Table 3 toxins-15-00684-t003:** Results from the two-way ANOVA and Tukey’s HSD test to assess pairwise differences between average congener relative abundance in each MC fraction for each experiment.

CR				
Congener	ANOVA *p-*Value	Pairwise Relationship	Difference (%)	Tukey’s Test *p-*Value
LA	ns	-	-	-
LR	0.02	SPATT-dMC	16.6	0.03
RR	0.01	SPATT-dMC	−20.9	0.01
YR	0.01	SPATT-dMC	5.8	0.02
		SPATT-pMC	6.1	0.02
LL				
LA	ns	-	-	-
LR	0.02	SPATT-dMC	45.0	0.004
		SPATT-pMC	35.7	0.02
RR	0.01	SPATT-pMC	−34.3	0.02
YR	0.01	pMC-dMC	−26.6	0.04

## Data Availability

The data presented in this study are available on request from the corresponding author.

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
