# Peer review of "Microcystin Concentrations, Partitioning, and Structural Composition during Active Growth and Decline: A Laboratory Study"

_toxins, 2023, doi:10.3390/toxins15120684_

Round 1
Reviewer 1 Report
Comments and Suggestions for Authors
Dear Authors,
Check the relevance of including a reference associated with the statement: ‘not all cyanobacteria are capable of toxin production and those that are, may not produce toxins continually.’.
Is it possible to standardize table 2 with regard to the number of significant numbers?
It is suggested that the references (line) be checked: 20 (442); 23 (450); 24 (452); 26 (457, 458); 28 (462); 32 (474); 33 (475, 477); 35 (482, 483); 37 (487, 488); 38 (491); 39 (493); 42 (501); 43 (502); 50 (519); 52 (523); 58 (536); 61 (542); , with regard to the standardization of their presentation.
Reviewer 2 Report
Comments and Suggestions for Authors
The manuscript titled “Microcystin Concentrations, Partitioning, and Structural Composition during Active Growth and Decline: A Laboratory Study” is a scientific work where the authors assessed the abundance of microcystin congeners according the time (in days) and the examined phases (particulate, dissolved and solid phase adsorption toxin tracking, respectively). The study is well-designed.
However, it exists some points that need to be addressed (please, see them below detailed point-by-point). The most relevant outcomes found by the authors can contribute in the growth of many fields like the development of better prognosis tools to forecast the blooms of cyanobacteria considering multiple environmental factors. For this reason, I will recommend the present scientific manuscript for further publication in Toxins once all the below described suggestions will be properly fixed.
Here, there exists some points that must be covered in order to improve the scientific quality of the manuscript paper:
1) INTRODUCTION. “MCs are the most commonly occurring cyanotoxins (…) investigated in various types of waterbodies including rivers, lakes, and estuares prone to cyanobacterial harmful algal blooms (cHABs)” (lines 29-32). The authors should also indicate the possibility to find the presence of microcystin in non-common environmental water bodies like cold permafrost areas [1].
[1] Gabyshev, V.A.; et al. Year-Round Presence of Microcystins and Toxin-Producing Microcystis in the Water Column and Ice Cover of a Eutrophic Lake Located in the Continuous Permafrost Zone (Yakutia, Russia). Toxins 2023, 15, 467. https://doi.org/10.3390/toxins15070467.
2) “The most common pathways for exposure (…) ingestion (…) contaminated fish, shellfish or drinking water are consumed” (lines 33-35). I agree with this information furnished by the authors but it is neccesary to take into account that water contamination like the presence of certain metals can induce the formation and growth of microcystin strains [2].
[2] Ceballos-Laita, L.; et al. Microcystin-LR Binds Iron, and Iron Promotes Self-Assembly. Environ. Sci. Technol. 2017, 51, 4841-4850. https://doi.org/10.1021/acs.est.6b05939.
3) “The World Health Organization (WHO) has published a set of guidelines (…) cyanobacteria cell densities (cell mL-1) and chlorophyll-a (chl-a) concentration (µg L-1)” (lines 41-43). The authors should provide some additional quantitative information about the recommended threshoulds by the WHO.
4) RESULTS. “After initial nutrient amendments (…) average chl-a levels peaking at 14,105 µg L-1 (…), 1,0023 µg L-1 (…) Charlton Pier” (lines 84-88). Please, the authors should homogenize the significant figures. This comment should be taken into account for the rest of the main manuscript body text (e.g. Table 2 in line 141).
5) Figure 1 (line 97). Why did the authors not homogenize the values concerning the X-axis and Y-axis among the data displayed in the different panels? This could aid to the potential readers to better compared between the tested conditions. Same comment for the Figure 3 (line 118).
6) DISCUSSION. “While no community data was collected to characterize (…) bloom-associated bacteria using high-throughput sequencing highlighted (…) upon bloom collections” (lines 213-217). Here, the authors need to list complementary ultrasensitive techniques as biomonitoring [3], single-molecule tools [4] or combination of mass spectrometry with enzyme linked immunosorbent assays [5] used for toxin detection.
[3] da Ferrão-Filho, A.; et al. Biomonitoring of cyanotoxins in two tropical reservoirs by cladoceran toxicity bioassays. Ecotoxicol. Environ. Saf. 2009, 72, 479-489. https://doi.org/10.1016/j.ecoenv.2008.02.002.
[4] Marcuello, C. Present and future opportunities in the use of atomic force microscopy to address the physico-chemical properties of aquatic ecosystems at the nanoscale level. Int. Aquat. Res. 2022, 14, 231-240. https://doi.org/10.22034/IAR.2022.1965012.1317.
[5] Badagian, N.; et al. Determination of Microcystins in Fish Tissue by ELISA and MALDI-TOF MS Using a Highly Specific Single Domain Antibody. Toxins 2023, 15, 84. https://doi.org/10.3390/toxins15020084.
7) CONCLUSIONS. This section is clear and concise. The authors highlight the most relevant outcomes found in this work. (OPTIONAL) Maybe it would be interesting if the authors outline the potential future avenues of this work.
8) MATERIALS & METHODS. “Natural bloom assemblages (…) PC carboys or PETG sampling bottles (…)” (lines 324-325). Please, the authors should add the full-names of “polycarbonate” and “polyethylene terephthalate glycol”. Then, the abbreviations should be placed between brackets.
9) “Samples (1-10 mL) were filtered onto 0.7 µm Whatman GF/Fs (…) to measure dMCs” (lines 348-349). Please, the authors should detail the name and country of all suppliers of the consumables and techniques used in this work. This point should be covered for the rest of this section.
10) REFERENCES. The references are mostly in the proper format style of Toxins. The journal name should appear in abbreviated form in some references (e.g. citations number 43, 45, 49, among many others).
Comments on the Quality of English LanguageThe authors should recheck the English in order to fix final details susceptible to be improved. The manuscript is generally well-written.
Reviewer 3 Report
Comments and Suggestions for Authors
The researchers conducted a lab study to determine the effects of time on both different congeners and phases of the liver toxin microcystin, produced by cyanobacterial Harmful Algal Blooms (cHABs) in samples taken from the Chowan River in North Carolina. Important findings of their lab experiments were that 1) particulate microcystin (pMC) is released from cells at exponential rates not related to chl-a or overall maximum microcystin concentrations and 2) dissolved microcystin (dMC) is more persistent than pMC for naturally occurring congener mixtures. In addition, the researchers went through the relevant literature and assembled Table 4, which is especially informative and contextualizes this studies’ results with respect to others’ findings on dMC and pMC half lives. Finally, the finding that the Solid Phase Adsorption Toxin Tracking (SPATT) bags may not give the same results as discrete dissolved samples is an important finding for those who wish to use these techniques as an easier way to monitor microcystins in situ. This manuscript is well written, cites appropriate sources, and adds new information to the growing body of knowledge on microcystin congener compositional change over time. After the minor changes I suggest below are made this manuscript should be accepted for publication.
Line 28- planktonic Anabaena has been renamed Dolichospermum. Please fix.
38- Replace “coastal” with “cultural”
Fig. 3- caption- do you mean “bar” rather than “column?”
Line 126-136- These are methods and should be moved to the Methods section.
Table 3- should the second pairwise comparison under RR be “SPATT-pMC?”
Lines 201,216,301- “Microcystis” should be italicized
Line 300- “Lake Eerie” should be replaced with “Lake Erie”
